# Orphan GPR26 Counteracts Early Phases of Hyperglycemia-Mediated Monocyte Activation and Is Suppressed in Diabetic Patients

**DOI:** 10.3390/biomedicines10071736

**Published:** 2022-07-19

**Authors:** Zahra Abedi Kichi, Lucia Natarelli, Saeed Sadeghian, Mohammad ali Boroumand, Mehrdad Behmanesh, Christian Weber

**Affiliations:** 1Department of Genetics, Faculty of Biological Sciences, Tarbiat Modares University, Tehran P.O. Box 14115-111, Iran; zahra.abedi@med.uni-muenchen.de; 2Institute for Cardiovascular Prevention (IPEK), Ludwig-Maximillians University, 80336 Munich, Germany; lucia.natarelli@med.uni-muenchen.de; 3Department of Electrophysiology, Tehran Heart Center, Cardiovascular Diseases Research Institute, Tehran University of Medical Sciences, Tehran P.O. Box 14155-6559, Iran; ssadeghian@tums.ac.ir; 4Department of Pathology and Clinical Laboratory, Tehran Heart Center, Tehran University of Medical Sciences, Tehran P.O. Box 14155-6559, Iran; mabroumand@yahoo.com

**Keywords:** diabetes mellitus, GPCRs, GPR26, monocytes, peripheral blood cells, hyperglycemia

## Abstract

Diabetes is the ninth leading cause of death, with an estimated 1.5 million deaths worldwide. Type 2 diabetes (T2D) results from the body’s ineffective use of insulin and is largely the result of excess body weight and physical inactivity. T2D increases the risk of cardiovascular diseases, retinopathy, and kidney failure by two-to three-fold. Hyperglycemia, as a hallmark of diabetes, acts as a potent stimulator of inflammatory condition by activating endothelial cells and by dysregulating monocyte activation. G-protein couple receptors (GPCRs) can both exacerbate and promote inflammatory resolution. Genome-wide association studies (GWAS) indicate that GPCRs are differentially regulated in inflammatory and vessel cells from diabetic patients. However, most of these GPCRs are orphan receptors, for which the mechanism of action in diabetes is unknown. Our data indicated that orphan GPCR26 is downregulated in the PBMC isolated from T2D patients. In contrast, GPR26 was initially upregulated in human monocytes and PBMC treated with high glucose (HG) levels and then decreased upon chronic and prolonged HG exposure. GPR26 levels were decreased in T2D patients treated with insulin compared to non-insulin treated patients. Moreover, GPR26 inversely correlated with the BMI and the HbA1c of diabetic compared to non-diabetic patients. Knockdown of GPR26 enhanced monocyte ROS production, MAPK signaling, pro-inflammatory activation, monocyte adhesion to ECs, and enhanced the activity of Caspase 3, a pro-apoptotic molecule. The same mechanisms were activated by HG and exacerbated when GPR26 was knocked down. Hence, our data indicated that GPR26 is initially activated to protect monocytes from HG and is inhibited under chronic hyperglycemic conditions.

## 1. Introduction

*Diabetes mellitus* (DM) is a chronic, metabolic disease characterized by elevated levels of blood glucose (hyperglycaemia). Type 2 diabetes (T2D) is the most common type of diabetes, accounting for approximately 90% of all cases of DM [1]. Hyperglycaemia, a hallmark in T2D, affects the metabolic homeostasis of several organs, like kidney, pancreas, and eyes, and induces vascular cell dysfunction and associated vascular diseases, including cardiovascular diseases (CVD), diabetic nephropathy, and neuropathies [2,3].

Inflammation plays a crucial role in the pathogenesis of T2D and its complications. Vascular cells, especially immune cells, such as lymphocytes and monocytes, are involved in the development of vascular inflammation, and promote endothelial activation and monocyte adhesion to the endothelium, sustaining diabetic vascular complications [2,4,5]. Despite data on the involvement of neutrophils and blood monocytes in diabetic patients still being contradictory, increased glucose uptake by monocytes from diabetic patients induces monocyte activation and enhances abnormal leukocyte–endothelial interaction [6,7]. Moreover, monocytes from diabetic patients are characterized by a moderate decrease in the activity of enzymes involved in the modulation of monocyte metabolism and inflammatory activation [8]. Mechanistically, high glucose (HG) levels, deriving from diabetes-associated chronic hyperglycaemia, induce inflammation via a variety of mechanisms, including ROS accumulation, pro-inflammatory MAPK cascade signalling, and NF-κB activation [9], which promotes the transcription of pro-inflammatory cytokines and adhesion molecules and related monocyte activation [6,10,11]. Moreover, HG can regulate monocyte apoptosis, although the role of this programmed cell death in diabetic patients is still controversial [12,13].

Cellular receptors are fine-tuned regulators of cell response to environmental changes via translating extracellular signals into intracellular activation cascades. The failure of a physiological process, such as the receptor-derived physiological processes in monocytes, dysregulates their function and mediates monocyte pro-inflammatory activation [3]. G-protein-coupled receptors (GPCRs) are the largest family of cell surface receptors in the human genome that play pivotal roles in a wide variety of physiological processes [14]. GPCRs emerged as novel receptors modulating the response of monocytes and other cells to environmental changes, such as hyperglycaemia [15]. Indeed, GPCRs can regulate the expression of cytokines and adhesion molecules to both exacerbate inflammation or promote its resolution [16,17]. Genome-wide association studies (GWAS) on diabetic patients indicate that GPCRs are differentially regulated in T2D patients and can influence the development and progression of T2D and vascular complications [17,18,19,20]. However, most of these GPCRs are orphan receptors, for which the mechanism of action in immune cells in diabetic patients is largely unknown [21]. Only a few studies exist on GPR119, free fatty acid receptor 1 (FFAR1/GPR40) and 4 (FFAR4/GPR120), and the bile acid receptor GPBAR1 (TGR5) as a developer of insulin resistance and *β*-cell dysfunction [22], receiving particular attention as targets for therapeutic interventions in diabetic patients [22]. Since orphan GPCRs emerged as the most differentially regulated sub-class of GPCRs in diabetic patients, the scientific community started to focus more on this sub-class of receptors [15,16,22]. Three orphan GPCRs have been reported so far to be involved in the development of diabetes and cardiovascular complications: GPR135 [19], GPR55 [23,24], and GPR39 [25]. However, data are still too preliminary and mainly focused on adipocyte-related diabetic complications [19,23,24,25]. Given the potential role of monocytes and orphan GPCR receptors in diabetes and associated cardiovascular complications, we investigated the role of GPCRs on monocytes from diabetic patients. Among almost 800 GPCRs existing, we focused on orphan GPCRs as the most important target receptor family for therapeutic purposes in biomedicine [21].

## 2. Materials and Methods

### 2.1. Bioinformatics Analysis of GWAS Available Databases

GWAS studies on single nucleotides polymorphisms and microarrays data from blood or peripheral blood mononuclear cells (PBMC) samples collected from diabetic patients (T1D and T2D) (GDS3963, GDS3874) were used to identify GPCRs differentially expressed and related to diabetes (Appendix A). Among all identified GPCR candidates, we excluded the T1D-related GPCRs and focused on those related to T2D. In the next step, we focused on orphan GPCRs [18,19,21,22]. Finally, GPR26 was selected as a T2D-related orphan GPCR. To predict functional associations between GPR26 and attributes (e.g., genes, proteins, diseases, etc.), we performed a bioinformatics analysis using the Harmonizome (2022 Database, Oxford) [26], JASPAR (9th release, V4.0, 2022) [27], KEGG (V 103.0, July 2022, Kanehisa Laboratories) [28], and STRING (V11.5, STRING Consortium 2022) databases [29].

### 2.2. Specimen Collection and Laboratory Investigations

Whole blood samples were obtained from 32 diabetic patients and 32 healthy donors referred to the outpatient clinic of Tehran Heart Center, Iran. The diagnosis of T2D was based on WHO criteria, and diabetic patients were selected based on standard criteria (fasting blood glucose (FBS)), N 125 mg/dL (6.9 mmol/L), and HbA1c N 6.5% (47 mmol/mol) (https://apps.who.int/iris/handle/10665/70523, accessed on 1 June 2022). To exclude any intervening condition due to background diseases, all of participants were examined for T1D, liver and kidney dysfunction, auto-immune diseases, current or prior cancers, acute or chronic inflammatory diseases, and any other unrelated disease. In addition, pregnant, smoking individuals and patients who were receiving immunosuppressive or hormone-containing drugs were excluded from the study. The healthy donors showed normal FBS and HbA1C values and did not report a T2D history, autoimmune disease, obesity, or any other chronic disease. This group was matched to diabetic cases based on gender, age, and ethnicity. Human PBMC were obtained from volunteers following an oral and written informed consent. Institutional approval was obtained from the Ethics Committees of Tarbiat Modares University (IR.MODARES.REC.1397.109). The study was conducted in accordance with the ethical guidelines of the Helsinki declaration.

### 2.3. Biochemical Analysis of Blood Samples from T2D and Healthy Donors

Whole blood samples were collected from precipitants after an overnight fasting. FBS was determined by glucose hexokinase method (Cobas Integra 400, Roche Diagnostics). HbA1c was quantified using an enzymatic method according to the manufacturer’s instructions (Cat no. DZ168A, Diazyme Laboratories, Inc., 12889 Gregg Ct. Poway, CA 92064 USA).

### 2.4. PBMC Isolation from Iranian Diabetic and Healthy Volunteers

PBMCs were isolated from fresh whole blood samples by density gradient using Ficoll Histopaque (10771, Sigma-Aldrich, St. Loius, MO, USA). In detail, 2 mL of Ficoll Histopaque per 1 mL of blood were placed in a 50 mL conical centrifuge tube. Fifteen mL of anticoagulated blood was diluted with an equal volume of Phosphate-buffered saline (PBS). The diluted blood was layered over the Ficoll Hypaque solution gently. Gradients were centrifuged at 400× *g* for 30 min at room temperature with no brake. Then, the PBMC layer was collected, washed with cold PBS, and centrifuged 10 min at 400× *g* (4 °C). The supernatant was discarded, and a wash step was repeated with PBS. PBMCs were finally collected and used for downstream analyses.

### 2.5. RNA Extraction and qPCR

Total RNA was extracted from PBMCs using TRIzol Reagent (15596026, Thermofisher Scientific, Waltham, MA, USA) according to the manufacturer’s instructions. One µg of RNA was reverse-transcribed with a high-capacity cDNA reverse transcription kit according to the manufacturer’s instruction (4368813, Thermofisher Scientific). Gene expression was evaluated by qPCR using the GoTaq qPCR master mix (A6001, Promega, Madison, WI, USA). All samples were tested at least in duplicate (technical replicates). The relative gene expression analysis of GPR26 was according to the Livak method [30]. Expression levels were normalized to B2M as stable internal control gene and expressed as 2^−ΔCt^. Primer efficiencies for the tested genes were comparable to those for reference genes. Primer sequences are reported in Appendix A.

### 2.6. Cell Culture and Treatments

PBMCs for cell culture experiments were obtained from additional 12 healthy volunteers after giving oral and written informed consent. Human monocyte cells (THP-1, ATCC, Rockville, MD, USA) were cultured in a Roswell Park Memorial Institute 1640 medium (RPMI 1640, 11530586, Gibco, Termofisher Scientific) supplemented with 2 mM L-glutamin, 1% penicillin/streptomycin, and 10% fetal bovine serum (Gibco), and incubated at 37 °C in a humidified atmosphere containing 5% CO_2_. CTHP-1 were exposed for 24, 48, and 72 h to media containing either 5 mM D-glucose (normal glucose, NG), or 25 mM D-glucose (high glucose, HG) to simulate a condition closed to the diabetic hyperglycemia. PMBCs were treated similarly for 24 and 48 h. Cell viability in each experiment was >90%, as indicated by trypan blue staining. THP-1 and PBMC total RNA was isolated as described above.

### 2.7. Antisense LNA GapmeRs Cell Transfection

THP-1 were chemically transfected (Lipofectamine RNAiMAX transfection reagent, 13778075, Life Technologies (Waltham, MA, USA), Termofisher Scientific) for 24, 48, and 72 h with LNA single-stranded antisense oligonucleotides (LNA-GapmeRs, 50 nM) designed for highly effective, strand-specific, knockdown of GPR26 mRNA, or with scrambled controls, alone or in combination with NG or HG. RNA isolation and cDNA synthesis were performed as mentioned before. Efficiency of GPR26 Gapmers was validated by qPCR and western blot. All data belong to at least three biological replicates.

### 2.8. Western Blot Analysis

Whole proteins were extracted from THP-1 using a RIPA lysis buffer (50 mM Tris pH 7.4, 150 mM NaCl, 1% Igepal, 0.5% sodium deoxycholate, 0.1% SDS) containing phosphatase and proteinase inhibitor cocktails (cOmplete Mini Tablets and PhosSTOP, 11836170001, Roche, Basel, Switzerland). The protein content was determined using the Pierce BCA Protein Assay Kit (23227, Thermo Fisher Scientific). Twenty µg of total proteins were loaded on a 4–12% precast polyacrylamide gel (NuPAGE™ 4 bis 12%, Bis-Tris, 1.0–1.5 mm, Mini-Protein-Gel, NP0321PK2, Invitrogen, Waltham, MA, USA) and electro-transferred onto polyvinylidene difluoride (PVDF) 0.45 µm membranes according to the manufacturer’s instructions. Proteins were incubated with antibodies indicated in the Appendix A. The β-actin and α-tubulin proteins were used for protein quantification and internal loading control.

### 2.9. Reactive Oxygen Species (ROS) Assay

To examine the possible role of GPR26 in generation of ROS, its amount was measured using the OxiSelect Intracellular ROS Assay Kit (STA-342, Cell Biolabs, San Diego, CA, USA) based on the manufacturer’s instructions. Briefly, THP-1 cells were transfected with GPR26 GapmeRs and exposed to NG or HG as indicated before. Then, cells were incubated with 2′, 7′-Dichlorodihydrofluorescin diacetate (DCFH-DA) during the last 30 min of treatment. At the end of the incubation, cells were washed three times with PBS to remove any DCFH-DA excess. Fluorescence was analyzed at 485 nm excitation/530 nm emission using a fluorescent plate reader (Infinite^®^ 200 PRO, Tecan, Männedorf, Switzerland). ROS levels were expressed as DCFH-DA relative fluorescence intensity and reported as fold change of the control.

### 2.10. Caspase-3 Activity Assay

THP-1 cells were cultured and transfected with GPR26 GapmeRs as mentioned above. To determine the activity of caspase-3, cells were incubated at 37 °C with a luminogenic caspase-3 substrate containing the tetrapeptide sequence DEVD (Caspase-Glo^®^ 3/7 Assay Systems, G8091, Promega). Luminescence is proportional to the amount of caspase activity present. Luminescence was measured in a plate luminometer (Infinite^®^ 200 PRO, Tecan). Six technical replicates per condition were performed for each of the 3 biologically independent experiments.

### 2.11. Monocyte Adhesion Using Calcein-AM

THP-1 were transfected with GPR26 GapmeRs for 24, 48 and 72 h. At the end of the experiment, cells were labeled with 5 μM/L × 10^6^ cells calcein-AM (Calcein, AM, Zellfarbstoff, C1430, Invitrogen, Life Technologies) for 45 min as previously described [31], with some modifications. Human Aortic Endothelial cells (HAoECs criopreserved, 500.000 cells, C-12271, PromoCell, Heidelberg, Germany) were cultured on gelatin-coated 96-well black plates. Confluent HAoEC monolayers were co-cultured with fluorescently labeled THP-1 cells for 20 min at 37 °C. Then, the monolayers were washed three times with PBS to remove non-adherent cells. Fluorescence was acquired at 492 nm (excitation) and 535 nm (emission) on a plate reader (Tecan). Relative fluorescence from treated monocytes adherent to HAoECs was normalized on that from the relative monocytes control groups. Comparisons were made between all four groups, as indicated in the Results. Representative images were acquired with a Thunder Imager Live Cell inverted microscope (Leica microsystem, Leica Mikrosysteme Vertrieb GmbH, Wetzlar, Germany) and equipped with a dual fluorescent and brightfield camera. Images were acquired for each of the 4 technical replicates and for all 3 biological replicates.

### 2.12. 3D Confocal Immunofluorescence for GPR26 Spatial Localization

THP-1 monocytes were plated on 8-well chamber slides (Ibidi) and treated as described before. Cells were fixed using the ViewRNA Cell Plus Assay (Fix & PERM™ Zellpermeabilisierungskit, GAS003, Affymetrix, Thermofisher Scientific), and incubated with antibodies against GPR26 (250 ng/mL dilution, NBP2-57693, R87210, Novusbio) and β-catenin (200 ng/mL, ab16051, Abcam). Proteins were detected using appropriate fluorescently labeled secondary antibodies feasible for 3D confocal microscopy, such as STAR-488-conjugated anti-rabbit IgG antibody (Molecular Probes, #A11034), and Cy™3 AffiniPure Donkey Anti-Rabbit IgG (H+L) (711-165-152, Jackson Immuno Research). Cells were embedded in Vectashield Antifade Mounting Medium with DAPI (H-1200-10, Vector Laboratories, Burlingame, CA, USA) to counterstain nuclei.

For detailed GPR26 localization studies, three-dimensional confocal laser scanning microscopy (CLSM) was performed with a Leica SP8 3X microscope equipped with a 63×/1.40 (Leica) oil immersion objective. Optical zoom was used where applicable. A UV laser (405 nm) was used for excitation of DAPI. A tunable white light laser for selective excitation of Star635P and Cy3 fluorochromes was used for the detection of GPR26 and β-catenin, respectively. Time gating of the detected emission signals (0.8–6.0 ns) was used for all STED channels to enhance image resolution. All data were acquired in three dimensions and voxel size was determined according to Nyquist sampling criterion. Image reconstructions were performed using the LAS X software package v.3.0.2 (Leica) and deconvolution was performed combining the Huygens Professional software package v.19.10 (Scientific Volume, Hilversum, The Netherlands) using the unsupervised CLSM algorithms.

GPR26 quantification and cell localization were performed using the Imaris 8.4.2 software equipped with the imaging processing toolbox MATLAB. Spatial distribution of automatically calculated GPR26 voxels (0.950 µm spot detection value) was defined according to the voxel localization to the cell surface, which corresponded to the membrane surface calculated using the spatial distribution of the β-catenin (0.289 µm smoothing, 3.46 µm threshold). A voxel value of 0.5 µm was used as threshold to discriminate membrane versus cytosolic GPR26. Multiple wells per experimental conditions were performed in independent experiments. The analysis of the staining was performed in a blinded manner.

### 2.13. Statistical Analysis

Statistical analysis was performed using GraphPad Prism 9.3.1 (GraphPad Software, Inc., San Diego, CA, USA). Data were initially evaluated for normal distribution with the Shapiro-Wilk test. Comparisons between two groups with normally distributed variables were analyzed by Student’s unpaired *t* test. Not normally distributed variables were compared by the Mann–Whitney test. Analysis of variance (ANOVA) was used to determine the significant differences between more than two groups. The correlation between the expression levels of GPR26 and the clinical features of patients were assessed by Spearman or Pearson tests based on the normality test result. In all cases, *p* values < 0.05 were considered significant.

### 2.14. Data and Resource Availability

The raw data are available in the Appendix A.

## 3. Results

### 3.1. GPCRs SNPs and Correlation with Their Expression in Diabetic Patients

*Diabetes mellitus* is a chronic, metabolic disease characterized by elevated blood glucose levels, which leads to additional chronic complications, such as cardiovascular, inflammatory, and kidney diseases [32]. Diabetes is among the top 10 leading causes of death worldwide, following a significant percentage increase of 70% since 2000 (WHO, https://www.who.int/news-room/fact-sheets/detail/the-top-10-causes-of-death, accessed on 1 June 2022). In Iran, diabetes is the 6th leading cause of death, with a death rate of 27.47 per 100.000 of population (WHO 2018 and World Health Rankings, https://www.worldlifeexpectancy.com/iran-diabetes-mellitus, accessed on 1 June 2022). Genome-wide association studies (GWAS) provided evidence that occurrence of certain single nucleotide polymorphisms (SNPs) in GPCRs are associated with an increased risk for diabetes and cardiovascular complications [33]. Hence, we analysed GWAS available datasets to depict all GPCRs with a reported SNP in diabetic patients. We identified 31 potential GPCRs candidates (Figure 1a). GPCRs are divided into six sub-classes based on their sequence and function. Accordingly, 15 out of 31 GPCRs were members of the Class A-rhodopsin-like receptors, 8 were members of the Class B-secretin family, 5 were members of the Class C-metabotropic of glutamate receptors, and 3 were members of the Frizzled or other GPCR subclasses [34] (Figure 1a).

Next, we analyzed the expression of the selected 31 GPCRs using available GEO datasets from blood samples isolated from T2D patients to define whether the presence of SNPs in the GPCR sequences correlated with a different expression of selected GPCRs in diabetic patients compared to healthy subjects. Twenty-nine out 31 GPCRs were differentially regulated in the blood or PBMC samples (Figure 1b,c). Next, we focused on orphan GPCRs as more attractive candidates for further studies. We end up with three orphan GPCRs, such as GPR35, GPR158, and GPR26. However, since GPR158 expression did not significantly change in the blood nor in the PBMC isolated from diabetic patients, and GPR35 is also known to be modulated in T1D, we end up with the orphan GPR26 as a promising candidate for our subsequent experiments (Figure 1b,c). Indeed, GPR26 was the most promising candidate among other orphan GPCRs, for which expression was decreased in the blood of T2D patients compared to healthy donors [33,35] (Figure 1c). GPR26 expression was slightly decreased in PBMC samples, probably due to the origin of the samples from young patients with an early T2D diagnosed disease.

To predict functional associations between orphan GPR26 and attributes, like genes and proteins associated with diabetes and related complications, we performed a bioinformatics analysis using the Harmonizome [26], JASPAR [27], KEGG [28], and STRING [29] databases. Harmonizome database results indicated that orphan GPR26 showed 1.763 functional associations with biological entities spanning 8 categories (molecular profile, organism, functional term, phrase or reference, disease, phenotype or trait, chemical, structural feature, cell line, cell type or tissue, gene, protein, or microRNA) extracted from 59 datasets (Appendix A). Analysis of functional associations predicted that orphan GPR26 might be involved in the regulation of cAMP catabolic processes (GO:0006198), vesicle formation and transport (GO:0071805), and of ADP-ribosylation factor (ARF) protein signal transduction (GO:0032012), which acts as regulator of phagocytosis and of immunity-related autophagy [36,37]. KEGG pathway analysis indicated that GPR26 might be involved in the cAMP, cGMP-PKG, insulin pancreatic secretion, MAPK, VEGF, oxidative phosphorylation, chemokine, mTOR, autophagy regulation, ABC transporters, T1D, and T2D pathways (Appendix A). Similar results were obtained using STRING, where G-protein coupled receptors, cyclic AMP-responsive element-binding proteins, and lysosome-related organelle complexes were predicted as functional partners of GPR26.

### 3.2. GPR26 Is down Regulated in the PBMC from Diabetic Patients and Negatively Correlates with Their BMI, HbA1c, and Insulin

GPR26 expression was first investigated in the blood collected from T2D and healthy Iranian donors. Demographic and clinical characteristics of control participants and patients are presented in Table 1. In detail, blood samples were collected from 32 patients (19 male, 13 females) with a diagnosed T2D and from 32 healthy donors (13 males, 19 females). None of the healthy donors were smokers or were hypertensive. Eleven diabetic patients got insulin routinely or had a history of using insulin. The diabetic and non-diabetic subjects did not significantly differ in gender, age, and body mass index (BMI) (Table 1). We did not observe sex-related differences within the diabetic or healthy donors in glycated hemoglobin (HbA1c), an index of long-term blood glucose concentrations, fasting blood glucose (FBS), nor on BMI. FBS and HbA1c were high in diabetic patients compared to healthy donors (Table 2). GPR26 expression was downregulated in diabetic patients compared to healthy donors (Figure 1d). We divided the diabetic patients according to their insulin supplementation (Table 2 and Figure 1d). In detail, 11 out of 32 T2D patients got insulin and mainly metformin + glibenclamide, an antidiabetic tablet medication, to lower the hyperglycemic levels and increase the insulin. The other 21 T2D patients did not get insulin but mainly the antidiabetic Glibenclamide medication (Table 2). Analysis of GPR26 expression in insulin versus non-insulin-treated patients indicated that GPR26 expression was decreased in patients taking insulin routinely or showing a history of using insulin compared to antidiabetic medication tablets treated patients, although the glycemic levels (HbA1c) were not different between the two groups (Table 3).

This data suggested that insulin-mediated alleviation of hyperglycemic adverse effects might counteract monocyte activation and GPR26 expression. Accordingly, we evaluated the association of *GPR26* expression with the glycemic profile, the HbA1c, and the BMI. *GPR26* was not correlated with FBS (Figure 1e) but showed a significant correlation with HbA1c (r = −0.38 *p* = 0.02) in diabetic patients (Figure 1f, Table 3). The association of *GPR26* with the BMI was also significant in both diabetic (r = −0.41 *p* = 0.01) and healthy donors (r = −0.35 *p* = 0.04) (Figure 1g, Table 3). Taken together, these data indicated that GPR26 is downregulated in diabetic patients and that insulin treatment might prevent GPR26 putative protective activation.

### 3.3. Hyperglycemia Upregulated GPR26 in PBMC and THP-1 Cultured Cells

We analyzed the expression of GPR26 in human cultured monocytes treated with different doses of glucose to simulate a hyperglycemic condition and to study GPR26 functional role. Accordingly, THP-1 cells were treated with 5 mmol/L (normal glucose, NG) or 25 mmol/L (high glucose, HG) of glucose for 24, 48, and 72 h. Interestingly, treatment of THP-1 with HG increased the expression of GPR26 (Figure 2a). The protein levels of GPR26 decreased over time when THP-1 cells were treated with HG, which increased GPR26 levels only at 24 and slightly at 48, but not at 72 h (Figure 2a). These data suggested that monocytes might response to hyperglycemic conditions by enhancing GPR26 expression, the translation of which into a protein is ineffective when cells are exposed to prolonged and chronic doses of HG.

To corroborate the data, human PBMC were isolated from healthy donors and treated with NG or HG for 24 and 48 h. Accordingly, we found that GPR26 expression was increased only at 24 but not at 48 h of HG exposure (Figure 2b). Accordingly, HG increased GPR26 protein levels at 24 but not at 48 h (Figure 2b).

Next, we analyzed GPR26 cellular localization and relative levels to test whether HG affected GPR26 exposure on the monocyte membrane surface. THP-1 were treated with NG or HG for 24, 48, and 72 h and stained for GPR26 and β-catenin as membrane marker. Immunofluorescence analysis 3D z-scan confocal images confirmed that HG increased whole GPR26 levels at 24, 48 but not at 72 h (Figure 2c). The number of voxels relative to GPR26 close to the β-catenin surface were reduced compared to that far from the surface, a value relative to the cytoplasmic GPR26 (Figure 2c). HG increased both cytoplasmic- and membrane-associated GPR26 levels at 24 and 48 h (Figure 2c). However, analysis of GPR26 localization at 72 h, a time point at which the GPR26 protein levels are no longer increased by HG, indicated that HG impaired GPR26 localization on the membrane surface of THP-1 and increased GPR26 cytoplasmic levels (Figure 2c).

Taken together, these data suggested that GPR26 might be activated to counteract the deleterious effects promoted by HG, which in turn impaired GPR26-related protective effect by inhibiting its membrane localization on the monocytes from T2D patients.

### 3.4. Knockdown of GPR26 Induced ROS Generation in THP1

To study the functional role of GPR26, THP-1 cells cultured in NG or HG were transfected for 24, 48, and 72 h with 50 nM of LNA single-stranded antisense oligonucleotides to knockdown GPR26, or with scrambled controls. GPR26 knockdown was confirmed by the analysis of GPR26 at mRNA and protein levels at all time points (Figure 3a). Transfection of THP-1 with LNA GapmeRs suppressed HG-mediated increase of GPR26 expression at levels lower than cells cultured with NG and transfected with scrambled controls (Figure 3a). Knockdown of GPR26 was confirmed at protein levels, in NG and HG treated THP-1 cells at 24 and 48 h (Figure 3a). No differences were visible at 72 h on GPR26 protein levels (Figure 3a). Immunofluorescence analysis confirmed that the main effect exerted by knockdown of GPR26 was in its cellular localization. Indeed, HG increased whole GPR26 levels at 24 and 48 h and decreased GPR26 at 72 h (Figure 3b). Knockdown of GPR26 inhibited the HG-mediated membrane localization and enhanced GPR26 cytoplasmic levels (Figure 3b). Taken together, these data suggested that GPR26 might be exposed on monocyte membrane to promote a mechanism of defense in monocytes, which is then inhibited by prolonged and chronic exposure to HG.

Accumulating evidence indicates that high levels of glucose promote dysfunctional monocyte activation, characterized by an increased inflammatory and apoptotic activity, partly by enhancing the production and accumulation of reactive oxygen species (ROS). To study the role of GPR26 on HG-mediated production of ROS, GPR26 was knocked down in THP-1 cells treated with NG or HG for 24, 48, and 72 h. Accordingly, we confirmed that HG increased ROS levels in HG-induced THP-1 cells compared to NG throughout the entire 72 h (Figure 3c). Knockdown of GPR26 exacerbated ROS production in a time dependent manner (Figure 3c), and enhanced HG-mediated ROS production (Figure 3c). Taken together, these data indicated that GPR26 played a protective role against the formation of free radical oxygen species, the production of which is enhanced by HG through the inhibition of GPR26 exposure on monocyte membrane and through GPR26 cytoplasmic localization.

### 3.5. Knockdown of GPR26 Increased ERK1/2 and p38 MAPK Activation in THP1 Cells

High levels of glucose can promote the activation of the MAPK signaling pathway, including p38 and ERK1/2, to promote monocytes-endothelial cell adhesion [38,39]. Moreover, recent findings indicate that ERK1/2 is regulated in response to the activation of various types of GPCRs [40,41], and that GPCRs mediate p38 MAPK activation to control monocyte inflammatory gene expression [42,43]. To explore whether GPR26 affected p38 and ERK1/2 activation, we assessed the levels of phosphorylated ERK1/2 and p38 by western blot, in THP-1 transfected with GPR26 GapmeRs, alone or in combination with NG or HG. Our results indicated that knockdown of GPR26 enhanced ERK1/2 and p38 MAPK activation in NG-cultured THP-1 at all time points (Figure 4a,b). ERK1/2 activation was increased in HG-treated THP-1 cells at 24 and 48 h. However, ERK1/2 activation persisted at 72 h only in HG-induced THP-1 cells where GPR26 was knocked down (Figure 4a).

Knockdown of GPR26 increased p38 MAPK activation at all time points (Figure 4b). P38 phosphorylation was promoted by HG only at 48 h, but remarkably induced at all time points when GPR26 was knocked down (Figure 4b). Taken together, these data indicate that GPR26 inhibits MAPK signaling pathways piloted by p38 and ERK1/2, independently from HG.

### 3.6. Knockdown of GPR26 Induced NF-κB p65 Activation and Monocyte Adhesion

Treatment of human monocytes with HG promotes the phosphorylation of p38 MAPK and ERK1/2, leading to NF-κB transactivation and related monocyte inflammatory activation [44]. GPRCs can regulate leukocyte activation and binding to the vessel wall through the endothelial-mediated expression of chemokines [45]. Despite controversy existing on the role of GPCRs as pro- or anti-inflammatory molecules, especially in diabetic patients, emerging evidence suggests that GPCRs and NF-κB might be connected in the regulation of inflammatory diseases [46].

To explore whether GPR26 affected NF-κB (p65) activation, THP-1 were transfected with GPR26 GapmeRs alone or in combination with NG or HG incubation [47]. Our results indicated that knockdown of GPR26 increased p65 expression at 24 and 48 h of NG (Figure 5a). HG and GPR26 knockdown showed similar effects on p65 activation (Figure 5a). However, p65 was enhanced at 72 h only when GPR26 was knocked down in HG-treated cells (Figure 5a). Hence, inhibition of GPR26 in diabetic patients might sustain hyperglycemia-mediated NF-κB activation.

Next, we examined the role of GPR26 on hyperglycemia-mediated monocyte inflammatory activation and adhesion to HAoECs. Accordingly, knockdown of GPR26 increased the adhesion of THP-1 to HAoECs at 48 and 72 h of NG (Figure 5b). HG-induced THP-1 adhesion to HAoECs was increased at all time points (Figure 5b). Knockdown of GPR26 enhanced the adhesion of THP-1 treated with HG compared to their respective HG-treated cells (Figure 5b). Moreover, knockdown of GPR26 enhanced the number of THP-1 cells adhering to HAoECs at all time points, especially of those co-treated with HG compared to NG (Figure 5b). Taken together, these data indicated that hyperglycemia promotes monocyte adhesion to ECs by suppressing GPR26′s anti-inflammatory role.

### 3.7. Knockdown of GPR26 Increased Caspase-3 Activity Promoted by HG

Several in vitro studies indicate that HG induces cell apoptosis in monocytes [48,49]. Caspase 3 plays a pivotal role in the apoptotic cell death process and therefore is widely used as a pro-apoptotic marker. Therefore, we investigated the effect of GPR26 knockdown, alone or in combination with HG, on THP-1 apoptosis by evaluating caspase 3 activation (cleavage of caspase 3 and its activity). Our data indicated that knockdown of GPR26 slightly affected Caspase 3 cleavage at 48 h (Figure 6a). Caspase 3 cleavage was increased when GPR26 was knocked down in THP-1 cells incubated with HG for 24 and 48 h, and the effect persisted even at 72 h (Figure 6a). HG alone significantly enhanced the ratio of cleaved caspase3 on total caspase 3 compared to NG at 24 and 48 h (Figure 6a). To confirm the functional activation of cleaved Caspase 3, we measured caspase-3 and -7 activity using an enzymatic assay. Our data indicated that knockdown of GPR26 increased Caspase 3/7 activity at levels comparable to that of HG at all time points (Figure 6b). Caspase 3/7 activity was significantly enhanced when GPR26 was knocked down in HG-treated cells at all time points (Figure 6b). These results confirmed that GPR26 plays an anti-apoptotic role and that GPR26 in monocytes might counteract HG-mediated increase of THP1 apoptosis. Indeed, inhibition of GPR26 increased the pro-apoptotic activity of caspase 3/7 mediated by HG after 72 h.

### 3.8. GPR26 down Regulation Significantly Inhibited Autophagy

Autophagy is an important physiological process activated to maintain cellular homeostasis and to adapt to extracellular cues. Autophagy is closely associated with ROS generation and inflammation [50]. Indeed, HG inhibits autophagy to promote ROS-mediated inflammasome activation and cytokine secretion in immune cells [50]. Microtubule-associated protein 1A/1B-light chain 3 (LC3) is a soluble protein that exists in the two forms of LC3-I and LC3-II, the second converted from LC3-I to initiate the formation and lengthening of the autophagosome. Measurement of LC3-II/LC3-I ratio consents the quantification of autophagy activation levels [51]. In addition, measurement of p62, a selective substrate of autophagy, is an additional method to measure autophagy, since it inversely correlates with the autophagy activation [52]. Thus, to explore the role of GPR26 in regulating autophagy in monocytes, we examined LC3-II/LC3-I ratio and p62 levels in THP-1 cells treated with NG or HG and transfected for 24, 48, and 72 h with GPR26 Gapmers. Our results indicated that knockdown of GPR26 decreased LC3-II/LC3-I ratio at 24 and 48 h (Figure 7a). HG decreased LC3-II at all time points compared to NG treated cells (Figure 7a). A significant decrease in LC3-II/LC3-I ratio was detected in THP-1, knocked down for GPR26, and treated with HG at 24, 48, and 72 h, a time point at which only the knockdown of GPR26 in combination with HG showed a significant effect on LC3-II decrease (Figure 7a). Next, we analyzed the levels of p62, the levels of which should inversely correlate with that of LC3-II. Accordingly, knockdown of GPR26 increased p62 levels in THP-1 cultured in NG at 24 h and 48 h (Figure 7b). HG increased p62 levels only at 48 h (Figure 7b). Knockdown of GPR26 enhanced p62 levels in HG-treated THP-1 at 24, 48, and 72 h, a time point at which neither GPR26 knockdown nor HG increased p62 levels (Figure 7b). Taken together, these data indicated that GPR26 promotes autophagy and might counteract HG-mediated autophagy inhibition. Hence, hyperglycemia might impair GPR26 to inhibit autophagy in monocytes.

## 4. Discussion

Diabetes is among the top 10 leading causes of death worldwide (WHO, https://www.who.int/news-room/fact-sheets/detail/the-top-10-causes-of-death, last accessed on 1 June 2022). T2D is one of the major public health diseases diffused in Iran, the Middle East’s second-largest country, due to its high prevalence rate [53]. Inflammation plays a crucial role in the pathogenesis of T2D and related complications [2,5]. The dysregulation of immune cells is strongly linked to T2D and vascular complications. Vascular inflammation is thought to be regulated by changes in monocyte and macrophage numbers, function, and related imbalance of pro-inflammatory signaling cascade [5]. Hyperglycemia acts as a strong activator of inflammatory response by activating endothelial cells and by dysregulating monocyte activation, leading to a chronic pro-inflammatory phenotype in monocytes and immune cells [2,5].

GPCRs play pivotal roles in a wide variety of immunological processes and are the target of nearly a third of all clinically utilized drugs [17]. Recently, several GPCRs involved in the development of insulin resistance and pancreatic β-cell dysfunction, such as GPR119, GPR146, GPR40, GPR120, and TGR5, have received attention as targets for therapeutic interventions in diabetic patients [54,55,56]. Although the mechanisms by which GPCRs regulate insulin sensitivity and immunological processes are unknown, the existence of GPCRs receptors as a very large and conserved family suggests that they might potentially modulate the response of immune cells in pathologies such as diabetes, atherosclerosis, and chronic inflammatory disease [17].

Accordingly, emerging studies on GPCRs, especially on orphan GPCRs, indicates their involvement in the regulation of immune cells biology and function [57]. For example, GPCR MRGPRX2 is expressed by monocytes to activate the intracellular pathways for an immunomodulatory action, leading to monocyte interaction with endothelial cells [58,59,60]. Our data indicated that orphan GPR26 is downregulated in diabetic patients and might be initially activated in monocytes and PBMCs to counteract the pro-inflammatory and apoptotic activation mediated by hyperglycemia. Chronic and prolonged exposure of monocytes to hyperglycemia impaired GPR26 protective effects and lead to its internalization and ineffective activation. Despite GPR26 ligand and mechanism of action needing to be further investigated, it emerged as a promising target to develop therapeutic strategies against inflammatory and diabetic diseases.

Analysis of GWAS studies from diabetic patients indicated that occurrence of certain SNPs in GPCRs are associated with an increased risk for diabetes and cardiovascular complications [33]. In addition, recent studies indicated that certain GPCRs are differentially regulated in diabetic patients [17,18,19,20]. However, the majority of these GPCRs are orphan receptors, for which the mechanism of action in diabetes and related complications is still unknown. In this study, we used GWAS studies to identify potential GPCR candidates with diabetes-associated SNPs and differentially expressed in diabetic samples. We selected orphan GPR26 as a promising candidate, presenting certain SNPs and downregulated in the blood samples from T2D patients. GPR26 is a central orphan GPCR that is most closely related to the serotonin receptor 5-HT5A and gastrin releasing hormone BB2 receptor, suggesting a possible role in the regulation of energy homeostasis and cell metabolism [61,62]. So far, the knowledge we have on GPR26 role belong to studies from *Caenorabditis elegans*, where depletion of GPR26 increases fat storage, and from rodents, where GPR26 deficiency in the hypothalamus increases genetic susceptibility to the onset of obesity, including glucose intolerance, hyperinsulinemia, and dyslipidemia (61). However, data regarding the role of GPR26 in diabetes and inflammatory associated complications are missing. We analyzed the expression of orphan GPR26 in the PBMC isolated from T2D patients and confirmed that GPR26 was downregulated. However, GPR26 expression was initially enhanced in human monocytes (THP-1) and human PBMCs treated with HG, a surrogate of hyperglycemia, whereas it was decreased when cells were exposed to prolonged and chronic HG levels. Two hypotheses were generated to explain these results. First, the group of diabetic patients was composed of insulin and non-insulin treated patients, which might have influenced the expression of GPR26. Second, GPR26 might be initially activated to protect monocytes against HG, whereas prolonged and chronic monocyte exposure to HG might impair GPR26 protective role. Our data indicated that both hypotheses might be reliable. Indeed, when diabetic patients were divided according to their insulin treatment, we found that GPR26 expression was further downregulated in insulin patients compared to non-insulin treated patients. This data might be due to insulin-mediated alleviation of hyperglycemic adverse effects on monocyte activation. In line with this, we noticed that the expression of GPR26 negatively correlated with the BMI and HbA1c percentage in diabetic and non-diabetic subjects. In addition, GPR26 exposure on monocyte membranes was decreased by chronic levels of HG, whereas GPR26 accumulated in the cytoplasm. Hence, these data supported the hypothesis that GPR26 might be activated to protect monocytes against hyperglycemia-mediated detrimental effects.

In the diabetic state, chronic or intermittent hyperglycemia promotes a phenotype named “glucose toxicity”, which is often associated with an imbalance of the cellular redox state, inflammation, and related apoptosis [63]. Indeed, several signaling pathways are altered and promote the formation of ROS, as well as the secretion of pro-inflammatory cytokines and cellular death (pathological autophagy and/or apoptosis). ROS plays a pivotal roles in triggering all these diabetic complications [63]. Accordingly, we investigated the role of GPR26 against all these HG-related complications. Our data indicated that orphan GPR26 knockdown promoted ROS production, MAPK-related inflammatory activation, monocyte adhesion, and apoptosis. Autophagy, commonly activated in immune cells to regulate inflammation following oxidative injury in diabetes [64], was impaired by GPR26 knockdown. Hence, our data revealed that GPR26 tightly regulates all crucial pathways related to monocyte cell biology and metabolism. Our data indicated that GPR26 protected monocytes from HG-mediated increase of inflammation and ROS production.

Previous studies showed that elevated ROS production leads to p38 MAPK andERK1/2 activation, which in turn promotes NF-κB-mediated transcription of pro-inflammatory genes and monocyte activation [38,44]. Our data indicated that lack of GPR26 protection under chronic hyperglycemic conditions enhanced MAPK and NF-κB activation. Hence, these data supported our hypothesis about GPR26′s protective role against T2D.

ROS induced -p38 MAPK and ERK1/2 activation results in caspase 3 activation and apoptotic cell death [65], despite several in vitro studies indicating that HG induces cell apoptosis in monocytes [48,49], Contradictory data exist regarding the role of apoptosis in monocytes from diabetic patients [12,13]. Indeed, inhibition of monocyte apoptosis impedes wound healing in diabetic mice, but it also promotes the development of diabetic complications in T1D patients [12,13]. Our data indicated that knockdown of GPR26 increased apoptosis and enhanced HG-mediated Caspase 3 activation, suggesting that GPR26 plays an anti-apoptotic role against HG.

Autophagy is fundamental to maintain cell homeostasis, removal of ROS, and inflammation [50]. Our results provided evidence that orphan GPR26 might play a protective role against HG by activating autophagy. Moreover, we identified a new role for GPR26 in promoting autophagy that is independent from HG. Although we did not investigate the mechanism in detail, HG and GPR26 might influence autophagic flux partly through p38 MAPK.

To the best of our knowledge, this is the first work demonstrating the expression and role of orphan GPR26 in monocytes from diabetic patients. Although additional data are needed to prove the protective role of GPR26 against T2D in vivo, we started to depict possible signaling pathways differentially regulated by GPR26 in THP1, independently or not from HG.

Indeed, our study has some limitations. Expression assay of GPR26 in PBMCs of diabetic patients was performed in limited samples, and future studies with a large number of samples are needed to confirm our results. Moreover, analysis of GPR26 expression and function in other cell types related to diabetes and cardiovascular complications, such as endothelial cells that are closely related to immune cell activation in diabetic patients, could provide more information and better clarify the role of GPR26 in T2D. Since the ligands of GPR26 are still unknown, future studies devoted to their identification might offer the basis for modulating the expression and activation of GPR26 in T2D patients.

## 5. Conclusions

The results obtained in the present study defined a new role for GPR26 in human monocytes and PBMCs in T2D patients. Our data suggested that THP1 might initially activate a mechanism of defense in response to hyperglycemia, which is partly mediated by a positive feedback loop that involves GPR26 activation. The protective effect might be impaired by chronic and intermittent hyperglycemic conditions. Treatment of T2D patients with insulin might counteract HG-mediated pathological activation of monocytes, a phenomenon partly reflected from a reduction in GPR26 expression. Moreover, our data suggested that GPR26 exerts a protective role independently from HG by inhibiting pro-inflammatory monocyte activation, ROS production, and apoptosis.

## Figures and Tables

**Figure 1 biomedicines-10-01736-f001:**
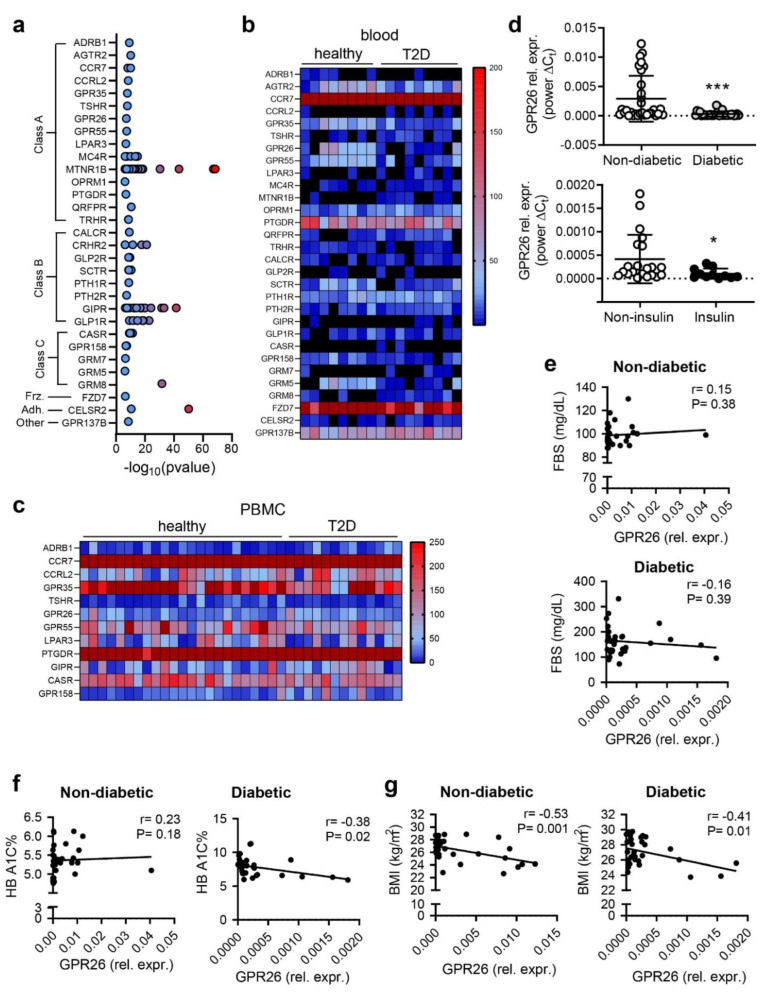
GPCRs candidates from GWAS, GEO datasets, and T2D patients. (**a**) GPCRs showing SNPs associated to diabetes. Analysis of available microarray GEO datasets and relative heatmap of 31 GPCRs differentially expressed (**b**) in the blood and (**c**) in the PBMC isolated from diabetic patients. (**d**) Expression analysis of GPR26 in diabetic patients (*n* = 32) in comparison to non-diabetic individuals (*n* = 32) (*** *p* < 0.001). (**e**) *GPR26* expression in patients treated with insulin (*n* = 11) compared to non-insulin treated patients (*n* = 21) (* *p* < 0.05). GPR26 mRNA expression was normalized to the corresponding B2M expression and expressed as power ∆Ct. (**e**–**g**) Correlation of GPR26 expression with (**e**) FBS, (**f**) HbA1c (%), and (**g**) BMI in non-diabetic and diabetic patients. Pearson correlation and r values are indicated.

**Figure 2 biomedicines-10-01736-f002:**
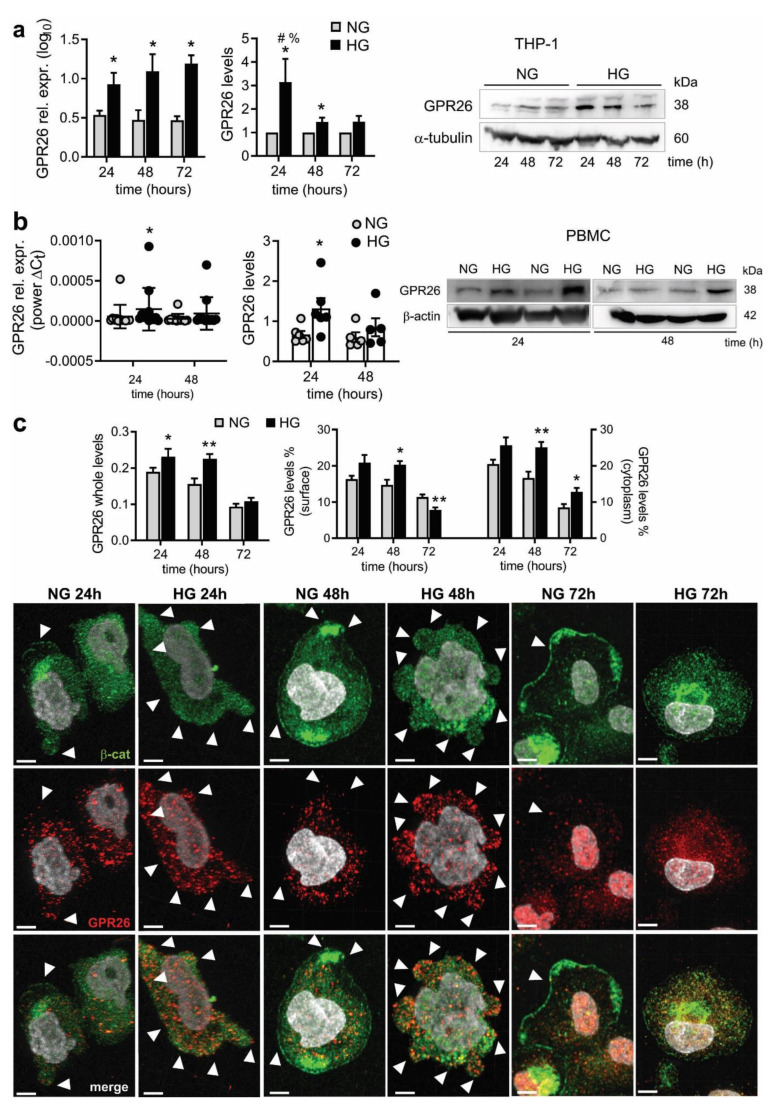
**Effect of HG on GPR26 mRNA and protein levels in human monocytes and cellular localization.** THP-1 and PBMC were treated with D-glucose 5.5 mM (normal glucose, NG) or 25 mM D-glucose (high glucose, HG) for 24, 48, and 72 h to analyze GPR26 in THP-1 (**a**) at mRNA (left) and protein (right) level. A representative Western blot (left) and relative expression of GPR26 (right). Densitometry plots report the fold change versus control. GPR26 intensity values were normalized to the corresponding β-tubulin value. GPR26 mRNA expression was normalized to the corresponding B2M expression and expressed as log10 (*n* = 4 independent experiments) (**b**) GPR26 mRNA (left) and protein (right) levels in PBMCs isolated from healthy donors and treated as indicated for THP-1. Densitometry plots reporting the fold change versus control. GPR26 intensity values were normalized to the corresponding β-actin value. GPR26 mRNA expression was normalized to the corresponding B2M expression and expressed as power ∆Ct (*n* = 4 independent experiments) (**c**) Analysis of GPR26 whole levels and cellular localization in THP-1 cells. Confocal 3D scanning images are representative of 3 independent experiments. White arrowhead indicates GPR26 localizing at the membrane surface. GPR26 levels were normalized to the cell surface, and on total number of cells (*n* = 3 ± SEM). Scale bar: 5 µm. * *p* < 0.05 ** *p* < 0.01 versus NG, # *p* < 0.05 48 h versus 24 h, % *p* < 0.05 72 h versus 24 h.

**Figure 3 biomedicines-10-01736-f003:**
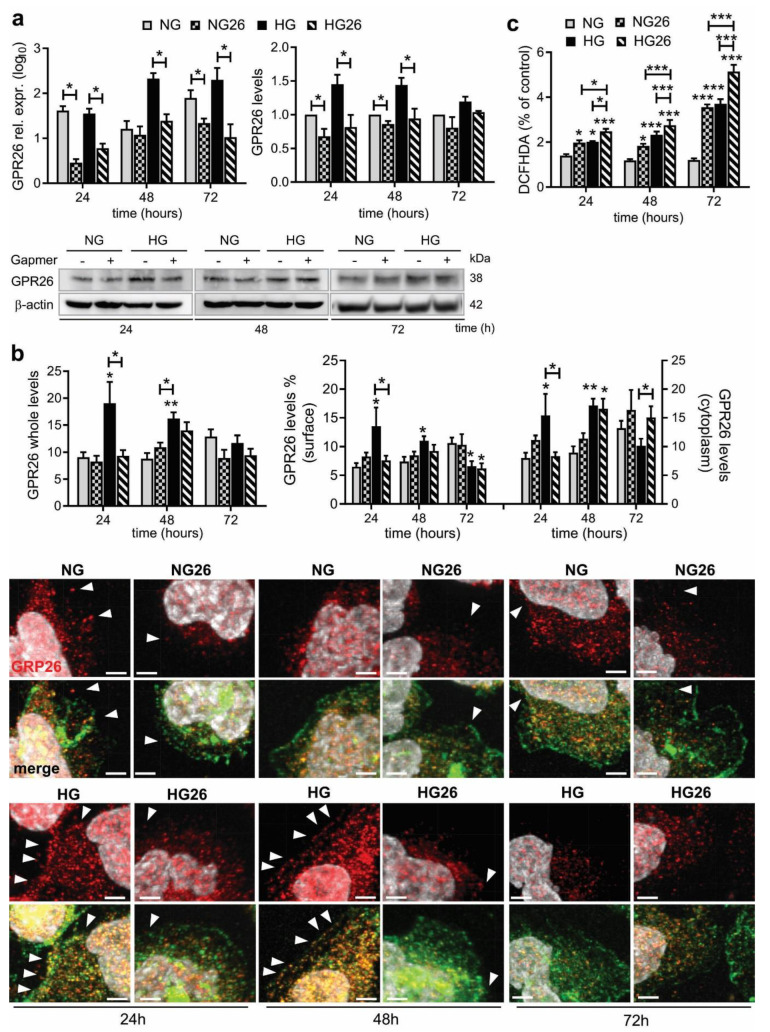
**GPR26 levels and localization in HG-treated THP-1 transfected with GPR26 Gapmers**. THP-1 were treated with NG or HG for 24, 48, and 72 h and transfected with GPR26 (+) or scrambled GapmeRs (−). (**a**) GPR26 mRNA expression was normalized to the corresponding B2M expression and expressed as relative expression (log_10_) (**left**). Densitometry plots reporting the fold change versus control. GPR26 intensity values were normalized to the corresponding β-actin value (**right**). (**b**) Confocal laser scanning microscopy images of THP-1 stained for GPR26 to analyze GPR26 whole levels, cytoplasmic, and membrane localization levels. White arrowhead indicates GPR26 localizing at the membrane surface. GPR26 levels were normalized to the cell surface, and then on total number of the cells. All values are represented as mean of at least 3 independent experiments (*n* = 3 ± SEM). (**c**) Intracellular ROS production. Data are expressed as percentage (%) of DCFH-DA fluorescence intensity relative to control. All values are represented as mean of at least 3 independent experiments. Scale bar: 5 µm. * *p* < 0.05; ** *p* < 0.01; *** *p* < 0.001 versus NG.

**Figure 4 biomedicines-10-01736-f004:**
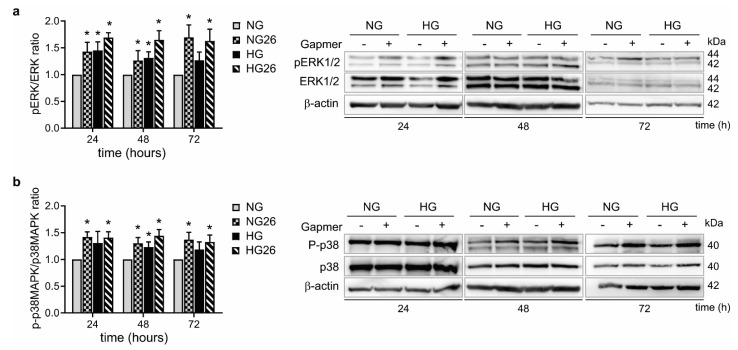
**The effect of GPR26 knockdown alone or in combination with HG on ERK1/2 and p38 MAPK activation**. THP-1 were treated and transfected for 24, 48, and 72 h with GPR26 (+) or scrambled GapmeRs (−), as indicated in the figures. (**a**) A representative Western blot of ERK1/2 and p-ERK1/2. Densitometry plot reporting the fold change versus control. ERK1/2 and p-ERK1/2 intensity values were normalized to the corresponding β-actin value and the ratio of p-ERK1/2 to ERK1/2 are presented as ERK1/2 activation. (**b**) A representative Western blot of p38 MAPK and p-p38 MAPK. Densitometry plot reporting the fold change versus control. p38 MAPK and p-p38 MAPK intensity values were normalized to the corresponding β-actin value and the ratio of p-p38 MAPK to p38 MAPK are presented as P38 MAPK activation. Images report one representative experiment out of at least 3 independent experiments. All values are represented as mean of at least 3 independent experiments (*n* = 3 ± SEM). * *p* < 0.05 versus NG).

**Figure 5 biomedicines-10-01736-f005:**
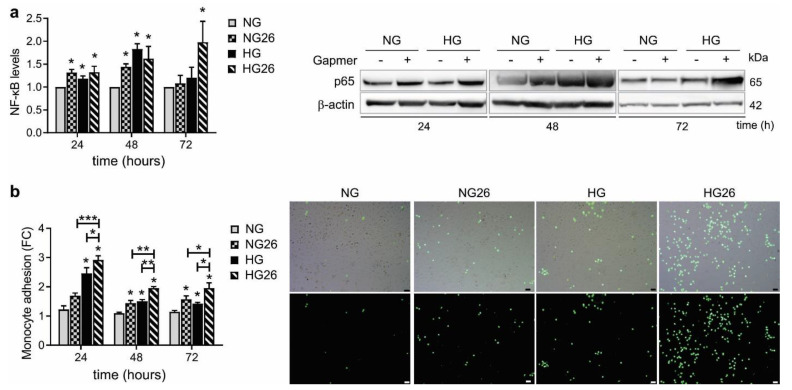
**The effect of GPR26 knockdown and HG on NF-kB and monocyte adhesion.** THP-1 were transfected with GPR26 gapmers (+) or scrambled GapmeRs (−) and treated with NG or HG to analyze NF-κβ (p65) activation and monocyte adhesion at 24, 48, and 72 h indicated in the figures. Panels show (**a**) a representative Western blot of NF-κβ. Densitometry plots reporting the fold change versus control. NF-κβ intensity values were normalized to the corresponding β-actin value. (**b**) Monocyte adhesion is reported as fold of change versus the basal adhesion of monocyte to HAoECs control (cells cultured in NG) that was set to 1. Right panels are fluorescence images of the monocytes, confirming that the small and bright spherical cells visible under microscope are indeed monocytes that were pre-labeled with Calcein AM. Images were taken using a 10X objective. All values are represented as mean of at least 3 independent experiments (*n* = 3 ± SEM). Scale bar: 50 µm. * *p* < 0.05; ** *p* < 0.01; *** *p* < 0.001 versus NG.

**Figure 6 biomedicines-10-01736-f006:**
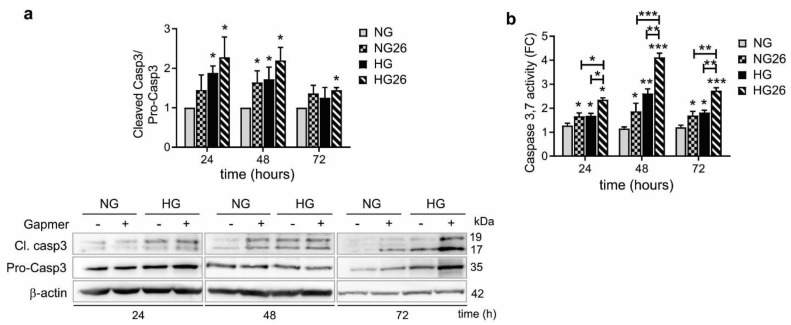
**The effect of GPR26 knockdown and HG on Caspase 3 activity.** The effect of GPR26 knockdown alone or in combination with HG on caspase activation at 24, 48, and 72 h with GPR26 (+) or scrambled GapmeRs (−), as indicated in the figures. (**a**) A representative Western blot of cleaved caspase-3 and total caspase-3. Densitometry plot reporting the fold change versus control. Cleaved caspase-3 and total caspase-3 intensity values were normalized to the corresponding β-actin value and the ratio of cleaved caspase-3 to total caspase-3 were presented as caspase activity. (**b**) Caspase 3,7 activity was measured using an enzymatic assay. Data are expressed as fold of change versus control (cells cultured in NG) that was set to 1. All values are represented as mean of at least 3 independent experiments (*n* = 3 ± SEM). (* *p* < 0.05 ** *p* < 0.01, *** *p* < 0.001 versus NG).

**Figure 7 biomedicines-10-01736-f007:**
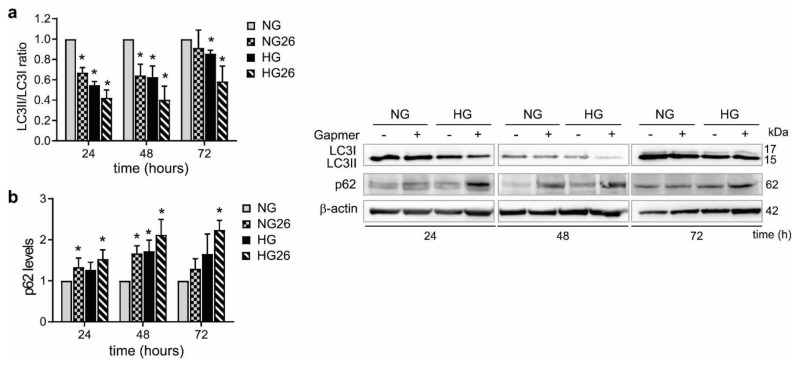
**The effect of GPR26 knockdown and HG on autophagy.** The effect of GPR26 knockdown alone or in combination with HG on autophagy at 24, 48, and 72 h with GPR26 (+) or scrambled GapmeRs (−), indicated in the figures. Panels show (**a**,**b**) a representative Western blot of LC3II, LC3I, and p62. Densitometry plot reporting the fold change versus control. LC3II, LC3I, and p62 intensity values were normalized to the corresponding β-actin value. The ratio of LC3II, LC3I were presented as the levels of activated autophagy. Images report one representative experiment out of at least 3 independent experiments. All values are represented as mean of at least 3 independent experiments (*n* = 3 ± SEM). * *p* < 0.05; versus NG.

**Table 1 biomedicines-10-01736-t001:** Demographic characteristics of the diabetic patients and healthy control.

Characteristics	Diabetic *n* = 32 (100%)	Non-Diabetic *n* = 32 (100%)	*p*-Value
Sex	19 males,13 females	13 males, 19 females	0.61 *^i^*
Age (years)	±55.75 ± 7.18	53.56 ± 4.85	0.15 *^ii^*
BMI (kg/m^2^) ^a^	25.98 ± 1.4	25.64 ± 1.29	0.32 *^i^*
Laboratory Plasma glucose ^b^ (mmol/L)	161.6 (73–331)	99.03 (88–130)	<0.0001
HbA1c ^c^	7.83 (5.92–11.29)	5.40 (4.76–6.3)	<0.0001
Glucose-lowering medications	11 (insulin), 21 (other medications)	-	-

^a^: Body mass index; ^b^: Low density lipoproteins; ^c^: Glycated hemoglobin; Data are presented as mean (SEM) for variables with normal distribution and median (interquartile range) for those without normal distribution. *^i^* Chi-square or Fisher’s exact test is performed to compare variables between diabetic and non-diabetic patients. *^ii^* Student’s *t*-test or Mann–Whitney *U* test is performed to compare variables between diabetic and non-diabetic patients.

**Table 2 biomedicines-10-01736-t002:** Glucose-lowering medications used to treat diabetic patients.

		Antidiabetic Medication Tablets	
T2D Patients [32]		Metformin Hydrochloride and Glibenclamide	Glibenclamide
Insulin	11	29	3
Non-insulin	21	3	29

**Table 3 biomedicines-10-01736-t003:** Correlation analysis between GPR26 mRNA expression and clinical characteristic of diabetic patients and non-diabetic individuals.

	Diabetic		Non-Diabetic	
Correlation with	r	*p*-Value	r	*p*-Value
HbA1c	−0.38	0.02	0.23	0.18
FBS	−0.16	0.39	0.	0.59
BMI	−0.41	0.01	−0.53	0.001

## Data Availability

C.W. and M.B. had full access to all data in the study and take responsibility for the integrity of data and accuracy of the data analysis.

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
