# Peer review of "Orphan GPR26 Counteracts Early Phases of Hyperglycemia-Mediated Monocyte Activation and Is Suppressed in Diabetic Patients"

_biomedicines, 2022, doi:10.3390/biomedicines10071736_

Round 1

Reviewer 1 Report

The manuscript by ZA Kichi et al. contributes do defining a role the so far novel, orphan GPCR member, GPR26 in human monocytes and PBMCs, in the context of Type 2 diabetes. The authors showed expression of GPR26 is negatively affected in PBMCs from subjects with T2D compared to healthy controls. By using GPR26 knockdown approach in human monocytes (THP-1 cells), the authors demonstrated significance of GPR26 in protecting monocytes against cell death. The manuscript holds novelty, is well written, however, before being further considered for publication in Biomedicines, the following minor and major comments should be addressed.

Minor comments:
1. Please remove any doubled parts of the manuscript (e.g. doubled "Material and Methods", lines: 238-392, "Results", lines: 395-401)
2. Abstract: line 37, please use (hyperglycemia (HG)), the right bracket is missing
3. What do the authors mean by "scientific committee", line 72?
4. Line 130- please correct "Thermos Scientific", line 614-please correct "controversial" to "controversy"
5. Figs. 2,3,5- what is the scale presented in the images?
6. Figs. 2,3- what is this white structure presented in fluorescent images?
7. Please carefully check the spelling and style, eg. line 92, should be "T2D-related", hyphen is missing
Major comments:
1. Please separate results and discussion-in the current version the results part contains some discussion-related comments, please include those in the "Discussion" section
2. Can the authors comment on sex-related differences in the obtained results, taking differences in groups (T2D vs. healthy donors) characteristics (men vs. women number), is there any sex-related effect, differences?
3. The authors show relatively broad range of HbA1c values in the analyzed population of subjects with Type 2 diabetes, 5.92-11.29, and also differences in the treatment used (Insulin vs. 21 other medications), can the authors discuss this aspect in more detail?

Author Response

The reply with detailed files is attached due to Tables and figures contained. We therefore made a PDF version of the point-by-point response to Reviewer #1

Reviewer 2 Report

Interesting review in the area of preclinical study of Orphan GPR26 counteracts early phases of hyperglycemia-mediated monocyte activation and is suppressed in diabetic patients.

Data on the involvement of neutrophils and blood monocytes in various pathological processes in patients with diabetes are quite contradictory. Some researchers note the absence of differences in the structure of neutrophil cytomembranes in healthy and sick people. Others— a statistically significant decrease in insulin-binding activity of blood mononuclear cells in pregnant women with type diabetes and gestational diabetes compared with healthy ones.

It is known that in monocytes with diabetes, all enzymes are characterized by a decrease in the number of cells with a high degree of reaction and a moderate decrease in the activity of all enzymes.

Most of these GPCRs are 21 orphan receptors, which mechanism of action in diabetes is unknown.

The data presented in the review indicated that GPR26 is initially activated to protect monocytes from HG, and is inhibited un- 31 der chronic hyperglycemic conditions.

The data obtained may lead to the development of new ways to prevent atherosclerotic complications in diabetic patients.

I don't have any questions or comments. The review of the manuscript biomedicines-1787030 can be published.

Author Response

We thank the reviewer for appreciating our work and for pointing out that the data regarding the role of neutrophils and blood monocytes in diabetic patients are still highly controversial. We believe that this work, together with the available literature regarding the presence of differentially regulated orphan GPCRs in diabetic patients may pave the way for new studies aimed at further investigating the role of orphan GPCRs as a link between diabetes and atherosclerotic complications.

In line with the suggestion to improve the introductory section, we have revisited the introduction and increased the references, including data on the still controversial and under-investigated role of circulating neutrophils and monocytes highlighted by the Reviewer.

Round 2

Reviewer 1 Report

The authors have addressed comments and improved quality of their manuscript.